Quantifying the scale dependence of primary productivity-species-richness relationships

Tavernia Brian G. btavernia@gmail.com
Open Space and Mountain Parks Department, City of Boulder , Boulder , CO , United States of America
White Easton
Electronic publication date: 2025 Nov 4
Publication date: 2025
Volume: 13
Electronic Location ID: e20297
Received 2025 Jul 12; Accepted 2025 Oct 6
Copyright: ©2025 Tavernia
Copyright year: 2025
Copyright holder: Tavernia
License: This is an open access article distributed under the terms of the Creative Commons Attribution License, which permits unrestricted use, distribution, reproduction and adaptation in any medium and for any purpose provided that it is properly attributed. For attribution, the original author(s), title, publication source (PeerJ) and either DOI or URL of the article must be cited.
License URL: https://creativecommons.org/licenses/by/4.0/

Keywords: Birds, Spatially varying coefficients, Bayesian statistics, Dynamic habitat indices, MODIS, Spatial autocorrelation, Breeding Bird Survey, Google Earth Engine, Remote sensing, Gross primary productivity

Funding: The author received no funding for this work. The funders had no role in study design, data collection and analysis, decision to publish, or preparation of the manuscript.

==============================
Vegetation productivity is expected to correlate with species richness, but there is debate about whether the relationship form (non-existent, negative, positive, unimodal) of productivity-species-richness relationships (PSRR) depends on the spatial extent and productivity measure used. Previous assessments employed coarse distance categories to examine scale dependence and did not consider scale dependence for alternative productivity measures. I used spatially varying coefficient models to precisely estimate the distances over which PSRRs change and to map spatial patterns of form for breeding birds across the conterminous United States. I created separate models for three measures summarizing intra-annual estimates of gross primary productivity: sum, minimum, and seasonality (coefficient of variation). Models demonstrated that PSRRs were scale-dependent, and PSRR relationships changed at median distances ranging from 1,010 to 2,184 km depending on the productivity measure. Previously used coarse distance categories would not have resolved the modeled distance estimates. Differences in median distance estimates across productivity measures were not statistically important. Across measures, PSRR form generally alternated between non-existence and positive, but there were pockets where seasonality negatively related to species richness in the western United States. While spatial patterns of form differed across measures, species richness in a small region of the western United States displayed a positive association with all three measures. Spatial patterns were related to prevailing productivity conditions. For example, sum tended to have a positive association with bird species richness in areas characterized by low annual productivity. This study novelly applies spatially varying coefficient models to address the long-debated scale-dependence of PSRR form, and the same approach is broadly applicable across geographies and taxonomic groups.

Introduction

Measures of ecosystem energy, including vegetation productivity, correlate with bird species richness at local to global scales (Bailey et al., 2004; Hobi et al., 2017). Some authors suggest that productivity-species-richness relationships (PSRR; Cusens et al., 2012) take different forms (non-existent, negative, positive, unimodal) depending on the geographic scale, or spatial extent, (Waide et al., 1999; Mittelbach et al., 2001; Phillips et al., 2010) although others argue that positive relationships predominate (Cusens et al., 2012). Knowledge of the precise relationship between geographic scale and PSRR form is limited by authors using pre-defined, coarse distance classes (Waide et al., 1999; Mittelbach et al., 2001; Cusens et al., 2012). For example, Waide et al. (1999) classified PSRRs as local (<20 km), landscape (20–200 km), regional (200–4,000 km), or continental-to-global (>4,000 km) when examining the prevalence of different forms. PSRR form is also hypothesized to depend on the ecological factor acting to limit species richness over a given geography. For North America, Phillips et al. (2010) found that PSRRs for birds were positive in areas with low average annual energy (presumably, energy-limited locations), non-existent in moderate-energy areas, and negative in high-energy areas. Phillips et al. (2010) used average annual plant productivity values, but there are other summary measures of productivity that may not all reflect the same limiting factors. For instance, intra-annual minimum productivity reflects environmental stress faced by organisms (Carrara & Vázquez, 2010). No one has yet precisely assessed how form varies with scale when productivity is characterized using different measures.

Spatially varying coefficient (SVC) models allow coefficient parameters to change across space, and these changes reflect differences in the magnitude and form of the relationship between a response variable and a covariate (Finley, 2011; Jarzyna et al., 2014). SVC models contrast with models assuming parameter stationarity, i.e., coefficient parameters remain the same everywhere. Applied to PSRRs, SVC models can characterize the relationship between geographic scale and PSRR parameters and identify regions associated with a particular form. Past researchers have used SVC models to study temporal turnover of bird communities (Jarzyna et al., 2014) and abundance trends (Meehan, Michel & Rue, 2019). These studies permitted parameters to change across discrete, coarse grid cells of approximately ten thousand (Jarzyna et al., 2014) or a million hectares (Meehan, Michel & Rue, 2019). Use of coarse grid cells potentially limits insight into the precise relationship between PSRR form and scale. Recently, researchers have demonstrated an approach that enables coefficient parameters in an SVC model to change over continuous space rather than across discrete areas (Zuur, Ieno & Anatoly, 2017; Davis, Croy & Snyder, 2024). This modeling approach estimates the distance, or range, within which coefficient parameter values are spatially correlated (i.e., share similar magnitude and form) and beyond which they are unrelated. Thus, SVC models based on a continuous space framework can estimate the precise geographic scale at which PSRR parameters change rather than relying on pre-defined distance categories. Output from the models can be used to map areas sharing relationship forms. SVC models with a continuous-space framework have not been applied to examine the relationship between scale and the form of PSRRs using alternative measures of vegetation productivity.

Researchers have defined a set of three dynamic habitat indices that capture separate aspects of plant productivity with each putatively linked to species richness through a different ecological mechanism (Radeloff et al., 2019). (1) Sum totals intra-annual vegetation productivity estimates for a location, and species richness is hypothesized to correlate positively with available energy. More available energy may increase population sizes and reduce extinction risks thus leading to greater species richness (Wright, 1983). (2) According to the environmental stress hypothesis, intra-annual lows in productivity act as bottlenecks restricting species richness, and bottleneck severity is assessed using minimum productivity. (3) Seasonality is measured using the coefficient of variation of intra-annual productivity estimates, and greater intra-annual productivity stability (i.e., less seasonality) is hypothesized to increase species richness. Researchers have found correlations between bird species richness and sum, minimum, and seasonality measures at regional, continental, and global scales (Coops, Wulder & Iwanicka, 2009; Coops et al., 2009; Hobi et al., 2017; Radeloff et al., 2019; Hobi et al., 2021). Despite these studies addressing separate scales, researchers have not evaluated how PSRR form changes across geographic scales when these different productivity measures are used.

Using SVC models with a continuous space framework, I examined the relationships between breeding bird species richness and sum, minimum, and seasonality plant productivity measures across the conterminous United States. Phillips et al. (2010) previously demonstrated that PSRR form varied across regions with low, moderate, or high average annual productivity, but they did not examine how PSRR form varies within a spatially continuous framework, nor did they consider productivity measures associated with different limiting ecological mechanisms. Consequently, I tested the following hypotheses: (1) the geographic scale at which PSRR form changes depends on the plant productivity measure used; (2) mapped patterns of PSRR form will differ across productivity measures; and (3) PSRR form will depend on whether productivity measures are relatively low, moderate, or high in a region. Importantly, this study does not seek to simply document the presence of scale dependence as has been done by previous studies. Rather, SVC models are used to precisely characterize scale dependence and map PSRR forms. This new approach to precisely quantify the relationship between PSRR and geographic scale is broadly applicable across geographies and taxonomic groups.

Materials & Methods

Study area

The study area was the conterminous United States (Fig. 1). Climate, physiography, and vegetation vary across this geography, and physiognomic vegetation types include boreal, hardwood, and mixed forests; rainforests; tallgrass, shortgrass, and mixed prairies; deserts; and brushlands (US NABCI Committee, 2000). In 2004, dominant covers (>10%) included grasslands (33.1%), savannas (19.6%), croplands (17.9%), and forests (15.9%); urban and built-up lands constituted 1.8% of the area. The preceding estimates come from the aggregation of International Geosphere-Biosphere Programme land cover classes described by Sulla-Menashe & Friedl (2022). Bird species richness and plant productivity measures for this geography are described below.

Figure 1 Average (2000–2004) estimated breeding bird species richness for North American Breeding Bird Survey route start points across the conterminous United States.

Bird species richness

Using data from the US Geological Survey’s Breeding Bird Survey (BBS; Sauer et al., 2017), I quantified bird species richness using data spanning the years 2000–2004 from 1,671 routes distributed across the conterminous United States. BBS routes are 39.4 km long and are placed in a stratified random manner along secondary and tertiary roads in the United States and Canada. Every May or June, an observer counts individuals for 3 min at each of 50 stops spaced at 0.8-km increments along a route. Observers limit their counts to a 400-m radius buffer of stops and begin their counts shortly after sunrise. Routes included in analyses had <50% coverage by croplands, urban and built-up lands, and cropland/natural vegetation mosaics within surrounding landscapes. Phillips et al. (2010) used this land-cover criterion to minimize human influence on PSRR models. Where possible in data and methods, I matched the approach of Phillips et al. (2010) because they documented a PSRR, and my interest lay in extending analyses of PSRRs by explicitly examining scale-dependence and alternative productivity measures. I represented landscapes as 39.4-km radius buffers of route start points and used 2004 International Geosphere-Biosphere Programme land cover types from the Moderate Resolution Imaging Spectroradiometer (MODIS) MCD12Q1 Version 6.1 product (Sulla-Menashe & Friedl, 2022). I aggregated 500-m land cover pixels to 1 km to match the resolution of productivity analyses (see below). Google Earth Engine (Gorelick et al., 2017) acted as the platform for landscape characterization. An additional filter retained routes only if their entire buffer fell within the boundaries of the conterminous United States. This filter ensured that productivity data were available for the whole landscape surrounding a route’s starting point.

Following Phillips et al. (2010), I did not include aquatic, exotic, raptor, or nocturnal bird species in richness estimates. Hydrology is likely more limiting for aquatic species than plant productivity, exotic species are dependent on anthropogenic habitats and disturbances removed via route filtering, and raptors and nocturnal species are inadequately sampled by the BBS. I considered aquatic bird species as belonging to the families listed in Kushlan et al. (2002) as well as other species in the orders Anseriformes and Charadriiformes and the family Oceanitidae. I identified exotic species according to Partners in Flight (2024), raptor species as defined by McClure et al. (2019), and species in the family Caprimulgidae as nocturnal. To account for heterogeneity in detection probability across species, routes, and years, I estimated species richness for each route and year using COMDYN software (Hines et al., 1999). For PSRRs, I modeled the average annual estimated species richness for routes.

Plant productivity measures

To calculate plant productivity measures, I used a 250-m resolution, 8-day gross primary productivity (GPP) data set created by Robinson et al. (2018) and available through Google Earth Engine (Gorelick et al., 2017). Phillips et al. (2010) found that 1-km resolution, average annual GPP correlated with mean species richness estimates of BBS routes. I aggregated the 250-m pixels from Robinson et al. (2018) to a 1-km resolution matching the spatial resolution used by Phillips et al. (2010). For each year, 2001 through 2004, I calculated the sum, minimum, and seasonality of monthly average GPP estimates per pixel in Google Earth Engine. Data for the year 2000 were unavailable. I averaged across years to produce the final, per-pixel estimates for each plant productivity measure. Estimates for each route represented the means of productivity measure values within 39.4-km radius buffers of route start points. Plant productivity measure estimates were standardized (i.e., mean centered and divided by standard deviation) prior to model fitting.

Statistical analyses

I modeled mean estimated species richness per route (Si) using an SVC model of the form: Si=αi+βiPMi+ɛi

where αi is a spatially varying intercept, βi is a spatially varying effect of a standardized productivity measure (PM), and ɛi is a residual. Separate models were created for each plant productivity measure. αi and βi are observations from a Gaussian Markov Random Field (GMRF) with a covariance matrix reflecting spatial autocorrelation among parameter locations. ɛi comes from a normal distribution with a mean equal to 0 and precision modeled using a log gamma prior distribution with a shape parameter equal to 1 and an inverse scale parameter set to 1 ×10−5. I plotted residuals versus fitted values and productivity measures to confirm an absence of residual patterns. I verified a lack of residual spatial autocorrelation through variograms.

A continuous domain stochastic partial differential equation (SPDE) supplied values necessary to parameterize a Matérn correlation function that supplied the GMRF’s covariance matrix (Zuur, Ieno & Anatoly, 2017; Davis, Croy & Snyder, 2024). The SPDE approach requires an irregular grid, or mesh, created through triangulation and covering the region to be modeled. To create the triangular mesh, BBS route starting points served as initial vertices with other vertices added so triangle edges did not exceed 449 km. This distance represented 10 percent of the maximum distance between BBS route start points. The mesh extended into a nonconvex hull surrounding BBS start points to avoid edge effects when estimating intercepts and standardized productivity measure effects for outer points. The mesh contained 3,166 vertices exceeding the 700–800 vertices recommended by Zuur, Ieno & Anatoly (2017) for initial analyses. I set penalized complexity prior distributions (Simpson et al., 2017) on SPDE parameters. Specifically, non-informative prior distributions specified a probability of 0.5 that the spatial range exceeded 2,243 km (half the maximum distance between BBS start points) and a probability of 0.5 that the marginal standard deviation exceeded 1.

To address the dependence of productivity measure parameters on geographic scale, I summarized Bayesian posterior distributions for spatial ranges and compared medians and 95% credible intervals across measures. To visualize the spatial pattern of PSRR forms, I projected median parameter estimates for plant productivity measures onto a 40-km resolution grid covering the conterminous United States. The 40-km grid cell size approximated the 39.4-km buffer established around BBS route start points. For each 40-km grid cell, I also determined whether the 95% Bayesian credible interval excluded zero, and, where it did, I considered the parameter to be statistically important. Finally, I evaluated evidence that PSRR form depended on the values of plant productivity measures. For 1-km raster grids representing productivity measures, I determined whether the relationship form at each grid cell center was non-existent (95% Bayesian credible interval overlapped zero), negative, or positive. Given that I had values for all grid cells (>34.1 million) not just a sample, I used boxplots to summarize the distribution of the plant productivity measures by form instead of employing inferential statistics.

I did not explicitly model the unimodal relationship form. Waide et al. (1999) suggested that the unimodal form may result from the accumulation of individual linear relationships (positive, non-existent, negative) along a productivity gradient. The SVC provides the potential for coefficient parameters to smoothly transition in magnitude and sign along a productivity gradient.

For all statistical analyses, I used R 4.5.0 (R Core Team, 2025). Fitting SVC models required the INLA 24.12.11 (Rue et al., 2017) and inlabru 2.12.0 (Bachl et al., 2019) R packages.

Results

Estimated bird richness ranged from 7.9 to 102.9 and averaged 56.3. Non-coastal, western routes had some of the lowest species richness values whereas high richness values were found across the study area (Fig. 1). The sum of plant productivity values extended from 0 to 19.1 kg*C/m2/year (median: 4.45) and was greatest in the eastern United States and along the west coast (Fig. 2). Minimum productivity was least restrictive in the southeastern United States and along the west coast, and overall values spanned from 0 to 1.1 kg*C/m2/month (median: 0.01). Generally, seasonality (coefficient of variation) increased moving from south to north and away from the west coast. Seasonality covered values from 0 to 3.3 (median: 0.80).

Figure 2 Average (2001–2004) productivity values from 1-km intra-annual gross primary productivity estimates derived from MODIS satellite data.

(Top) Sum values are the totals of intra-annual productivity estimates. (Middle) Minimum productivity values represent the lowest intra-annual productivity estimate. (Bottom) Seasonality is the coefficient of variation of intra-annual productivity estimates.

Sum, minimum, and seasonality parameters possessed median geographic scale estimates of 1,618 km (Bayesian 95% credible interval: 969–2,628 km); 2,184 km (1,356–3,634 km); and 1,010 km (577–1,722 km), respectively (Fig. 3). In one location in the southwest, a positive PSRR form occurred for all three measures whereas, across other large regions, there was an absence of statistically important PSRR forms regardless of the productivity metric modeled (Fig. 4). Statistically important and positive sum parameter estimates occurred across large portions of the western and northeastern conterminous United States; there was no effect of sum in other areas (Fig. 4). Areas where the sum of productivity had no effect on species richness possessed relatively high sums, whereas areas displaying a positive form had relatively low values (Fig. 5). Minimum plant productivity parameter estimates were statistically important and positive in the southwestern United States and not important elsewhere (Fig. 4). Minimum productivity exhibited non-existent and positive forms with a positive form being more likely in areas with relatively low minimum productivity values (Fig. 5). In contrast to sum and minimum plant productivity parameters, the seasonality parameter did take on a negative form in some locations. Specifically, statistically important seasonality parameter estimates occurred in smaller pockets within the western United States and displayed both positive and negative forms (Fig. 4). There was also a small area of the northeastern United States where seasonality positively correlated with species richness (Fig. 4). The non-existent form showed a wider spread of seasonality values compared to positive and negative forms (Fig. 5). A negative form was more likely in areas with relatively low seasonality values compared to areas with non-existent or positive forms.

Figure 3 Bayesian posterior distributions of spatial range parameters for models relating estimated breeding bird species richness to productivity measures.

The range represents the distance within which parameter estimates for productivity measures are spatially autocorrelated. Sum is the total of intra-annual productivity estimates, minimum productivity represents the lowest intra-annual productivity estimate, and seasonality is the coefficient of variation of intra-annual productivity estimates.

Figure 4 (Right) Spatially continuous estimates for parameters relating productivity measures to breeding bird species richness and (Left) whether parameter estimates were statistically important.

Estimates represent the median of Bayesian posterior distributions and were considered important where 95% credible intervals did not overlap zero. (Top) Sum values are the totals of intra-annual productivity estimates. (Middle) Minimum productivity values represent the lowest intra-annual productivity estimate. (Bottom) Seasonality is the coefficient of variation of intra-annual productivity estimates.

Figure 5 Boxplots showing the distribution of sum, minimum, and seasonality productivity measures by productivity-species-richness relationship form.

The boxplots are based on values observed at >34.1 million 1-km grid cells across the conterminous United States. Outlier values are not displayed on box plots. The negative form was not observed for sum and minimal productivity measures. (Top) Sum values are the totals of intra-annual productivity estimates. (Middle) Minimum productivity values represent the lowest intra-annual productivity estimate. (Bottom) Seasonality is the coefficient of variation of intra-annual productivity estimates.

Discussion

My analysis showed that it is important to accommodate non-stationary productivity-species-richness-relationships (PSRR) through appropriate statistical methods such as spatially varying coefficient (SVC) models permitting parameter estimates to change continuously across space or appropriately sized grid cells. Using SVC models with a spatially continuous framework, I estimated the geographic distance at which PSRR models change in contrast to previous researchers who used distance categories to investigate the relationship between PSRRs and scale (Waide et al., 1999; Mittelbach et al., 2001; Cusens et al., 2012). My approach aligns with previous work addressing the value of retaining continuous variables for ecological analyses rather than discretizing them (McGarigal, Tagil & Cushman, 2009; Beltran & Tarwater, 2024). My study area would have fallen into a single distance category used by previous researchers (regional: 200–4,000 km; Waide et al., 1999; Mittelbach et al., 2001; Cusens et al., 2012) and potentially led to the assumption that a single PSRR form applied across the area, an assumption opposing my finding of a non-stationary relationship. Large regions typically in the western United Stated displayed positive relationships between productivity measure parameters and bird species richness whereas other areas typically showed no relationship. Adopting this modeling approach revealed that, despite what might be expected based on fundamental ecological theory (Wright, 1983), productivity did not ubiquitously constrain species richness nor possess a scale-invariant relationship with species richness, an observation supporting the findings of some past reviews (Waide et al., 1999; Mittelbach et al., 2001).

The geographic scale at which PSRRs changed did not depend on the plant productivity measure, but mapped patterns of PSRR forms differed across measures. Assuming that productivity measures differ in mechanistic linkages with species richness and would have dissimilar spatial distributions, I predicted that PSRRs change at different geographic scales depending on the measure used. While median distance estimates differed, overlapping credible intervals did not provide strong statistical support for my prediction. The putative ecological linkages between species richness and measures act locally to determine the relative fitness of individuals and representation of species, and thus, there may not be a direct link between these mechanisms and the geographic scales at which PSRRs change. Even if a link exists, the mechanisms may not act equally across all constituents of the breeding bird community. For example, migratory species would not be directly subject to energetic bottlenecks. Spatial gradients in relative values differed across plant productivity measures; for example, the sum of plant productivity values increased from west to east whereas seasonality increased from south to north. These dissimilarities in spatial distribution among plant productivity measures were apparently not enough to generate statistically important differences in the relationship between PSRRs and geographic scales. Despite this, the relative values of plant productivity measures were associated with mapped PSRR forms.

Depending on the plant productivity measure, PSRR forms displayed different associations with the relative values of a measure. In general agreement with the findings of Phillips et al. (2010), the sum of intra-annual productivity showed a positive relationship with species richness in areas often characterized by low annual productivity, but, contrastingly, I did not find a negative form in areas characterized by high annual productivity. Instead of SVC models, Phillips et al. (2010) divided BBS routes into low, moderate, and high categories using equal interval divisions of average annual GPP estimates. Routes in a category were not necessarily spatially adjacent and could be separated by many intervening routes in a different category. They regressed breeding bird species richness against average annual GPP in each of their productivity categories to determine PSRR form. I used SVC models to allow the smooth, spatially continuous transition of sum parameter estimates and then examined the relationship between form and average annual productivity sum. The difference in methods is likely the basis for the contrasting results observed across the two studies. I argue that the SVC method is preferred as changes in PSRR form may reflect spatially autocorrelated ecological factors or processes, such as species movements among local communities.

PSRRs involving minimum productivity displayed a positive form in areas with relatively low minimum values whereas the relationship was often non-existent in areas with higher values. This observation may indicate that there is a minimum intra-annual productivity threshold above which increasing productivity is relatively inconsequential. Unlike PSRRs involving the other two plant productivity measures, PSRRs including the seasonality measure displayed negative in addition to non-existent and positive forms. While increased seasonality is expected to reduce species richness due to resource instability, at least some segments of breeding bird communities respond positively to increased seasonality (Hurlbert & Haskell, 2003). Relative to areas with non-existent seasonality forms, areas with positive or negative forms were characterized by values that were not extremely low or high. This suggests that factors other than seasonality are limiting bird communities in areas at the extremes of the seasonality spectrum. Increased seasonality was negatively associated with species richness in areas typified by relatively low seasonality. Gaining further insight into the observed patterns may require modeling the response of species richness to separate aspects of seasonality, namely amplitude, timing, and predictability (Hernández-Carrasco et al., 2025).

The three plant productivity measure were positively associated with species richness in an overlapping area of the western United States. As might be expected based on the observations for individual measures, over a third of this area is characterized by a combination of low sum, low minimum, and moderate seasonality values. Pairwise value combinations (e.g., low annual and minimum productivity values) that promote positive associations with two of the three measures are also common.

The middle and eastern United States possessed large areas lacking statistically important relationships between plant productivity measures and breeding bird species richness. Other studies using the three productivity measures have largely reported significant correlations with bird species richness (Coops et al., 2009; Hobi et al., 2017; Hobi et al., 2021), but methodological and analytical differences make direct comparisons difficult. Summarizing breeding bird species richness and productivity measures by United States ecoregions, Coops et al. (2009) reported that minimum plant productivity and coefficient of variation (seasonality) measures correlated with bird species richness, although there was no relationship with cumulative sum. Coops et al. (2009) calculated productivity measures using MODIS-derived fraction of absorbed photosynthetically active radiation rather than GPP. Using the approach of Coops et al. (2009), but deriving measures from MODIS-based GPP, Hobi et al. (2017) reported quadratic relationships between breeding bird species richness and the three productivity measures. Cumulative sum explained relatively little variation of richness. Quadratic models suggest there are sets of values over which the relationship between richness and productivity takes different forms, including being non-existent. Hobi et al. (2021) related breeding bird species richness to the mean and standard deviation of GPP-derived productivity measures within 39.4-km buffers surrounding Breeding Bird Survey route centroids in the conterminous United States. They reported negative relationships between species richness and mean minimum productivity and standard deviation of the coefficient of variation measure whereas all other parameters displayed positive associations. These studies did not employ methods, such as SVC models, that allow parameter estimates to vary across space and potentially reveal areas with non-existent PSRR forms.

Future studies could use SVC models to explore the correlations between richness and other environmental factors beyond plant productivity measures, such as climate and land cover, potentially affecting species richness. Of course, such studies need to consider whether climate and land cover metrics are correlated with productivity and would thus be unlikely to explain species richness where productivity measures have already failed to do so. Hobi et al. (2021) found that adding environmental and climate variables to productivity measures improved model performance. These variables may be of particular importance in areas lacking statistically important plant productivity effects.

Conclusions

My study demonstrates that PSRRs for breeding bird communities are scale-dependent, shifting in form across the conterminous United States. Using SVC models, I treated distance as continuous in these analyses and found that previously used distance categories are too imprecise to describe the relationship between PSRRs and geographic scale in this system. Spatial patterns of positive, negative, and non-existent forms differed depending on the productivity measure used. This met expectations as the spatial distributions of productivity measures were dissimilar and different mechanisms are hypothesized to link species richness to the alternative measures. Prevailing productivity conditions influenced the form of relationships. For example, the sum of intra-annual productivity positively influenced species richness in areas typified by low productivity. While spatial patterns of form differed, the distances over which PSRRs change did not differ statistically across productivity measures. A large portion of the study area lacked significant PSRRs, and future studies should consider additional variables, such as land cover and climate, in addition to productivity. The use of SVC models to precisely examine the scale-dependence of PSRRs is novel and can be applied to other taxonomic groups and geographies.

Supplemental Information

Supplemental Information 1 Script files and input data required for recreating models in the article

I thank Michael Reed, Allen Hunt, Easton White, and an anonymous reviewer for comments improving an earlier version of this manuscript.

Additional Information and Declarations

Competing Interests

Author Contributions

Data Availability

The author declares that they have no competing interests.

Brian G. Tavernia conceived and designed the experiments, performed the experiments, analyzed the data, prepared figures and/or tables, authored or reviewed drafts of the article, and approved the final draft.

The following information was supplied regarding data availability:

The code to recreate the models and input data are in the Supplemental File.

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
