# Peer review of "Quantifying the scale dependence of primary productivity-species-richness relationships"

_PeerJ, doi:10.7717/peerj.20297_

## Round 0.1 · original submission · Major Revisions

Both reviewers provide detailed feedback that should be addressed in a revision. The reviewers point out areas of the text that can be improved for clarity, justification of the approaches, literature that should be discussed and cited, and highlighting potential limitations and also importance of the work.

Reviewer 1 ·

Basic reporting

The language is clear, but I find that the reporting is confusing with respect to the content, and that the paper focuses on a single aspect (the structure of the model) with little attention to the underlying science and details.

This begins with the title. The paper really is about the relationship between breeding bird diversity and primary productivity (plants). The title does not mention birds, and it is therefore naturally understood as the dependence of primary productivity on the species richness of primary producers. Also, the title is misleading as it suggests that productivity (of plants) would depend on species richness (of birds), although, if the relationship is causal at all, it actually is the other way round.

I further find terminology to be rather vague. For example, the author uses the terms "ecosystem energy" and "annual energy", but these really are not precise terms. Would it be "energy content", such as e.g., calories released when the entire ecosystem (possibly including soils) is combusted? Or is it more "energy flow"? If yes, through which compartments? Reading the paper, I realized that gross primary productivity is used as a metric of "energy". However, the reported data cannot be correct. For example, productivity values (e.g., GPP, line 200) are reported as up to 190 tons of C per m2 and year. However, a typical forest will have GPP values in the range of maybe up to a few kg of C per m2 and year. The values presented simply do not make sense.

Experimental design

This is an observational study relating bird diversity observations to metrics of plant productivity. As far as the design and analyses are described, they seem to be correct.

I am not familiar with the linear models with spatially varying coefficients that are used here. This requires verification by a statistical expert.

The author basically fits bird species richness as a linear function of plant productivity, with coefficients that vary depending on spatial scale.

Linearity is a strong assumption, given that, as the authors report, these relationships can also be unimodal (e.g., stated on line 39). The model cannot accommodate that.

Validity of the findings

I would have liked to see a discussion of alternative explanations for the patterns observed.

For example, is plant primary productivity really the driver of bird species diversity? Or is it rather the presence of particular habitat types that is important for the birds, because these habitats are used as breeding grounds, or because particular bird species find the food they need there (e.g., particular seeds, insects, or other animals)? Also, usually areas with a diversity of habitats and ecotones (e.g., forest-grassland transitions) harbor the highest species diversity.

The work presented focuses on a methodological aspect, namely the statistical model and technique used to linearly relate bird species richness to plant productivity (i.e., a linear model with intercept and slope varying continuously with spatial extent). Apart from this, the paper seems to me like a sweeping generalization with very little focus on the underlying ecology and mechanisms.

Additional comments

Overall, I find that the paper provides little insight, apart from demonstrating the application of a model fitting technique that may or may not have certain advantages. Whether the results obtained in this way really are better than the results obtained by the existing analyses remains unclear to me.
The paper certainly should provide far more attention to the underlying ecology. It is also inaccurate in the use of technical terms (e.g., "energy"), and some values reported cannot be correct (e.g., GPP values).

·

Basic reporting

-

Experimental design

-

Validity of the findings

-

Additional comments

Line 158. All statistical inputs have means of zero, and their effects are additive. Even if they are correlated, I do not so easily see how a non-zero mean in species number can result. At least, this should be better explained.

Line 202. Minimum productivity was least restrictive along the West Coast and in the Southeast. This result is not surprising, although one might suspect something like north of Monterey Bay, where precipitation becomes more reliable. Thus, a relationship between minimum productivity and species number should be expected to be strongest where the climate is most variable (which automatically includes more arid regions).

Overall, it becomes difficult for me to see what the bases were for the various statistical inputs chosen. What specific problems in information gathering do they address, and what do we gain from them?
Further, I would have liked to see more (at the beginning already) about why environments with higher productivity might support more birds. Is it because there is more food to survive on, and larger populations support more individuals, with more individuals supporting more species? Or is this a two-layer argument? More productivity supports more plant species, and a greater number of plant species supports a larger number of specialist bird species, which use either the plants (or plant-specific insect grazers) as food sources. The results obtained do not lead me to any obvious conclusions regarding the actual means by which productivity could enhance species richness (and thus why it might not be correlated in some regions – e.g., lack of suitable habitat, prohibitive temperature and water extremes, or because the climate seasonality is unsuited for some migrating species, etc). But this relates ultimately also back to the original hypothesis, namely that changes in scale could influence the form of the PSRR. Of course, they could. But how and why? The differences between east and west, for example, could relate to the much larger variability of topography on smaller scales in the west than in the east of the USA. Currie (1991) noted that the relationship between productivity (as measured in evapotranspiration) tended to underestimate plant species number in areas of high heterogeneity, such as California. This could possibly be accounted for, for example, by looking at the scale dependence of the species number and seeing how the distinction between similar ET values in the east and west provided an understanding of the relevance of topographic (and climatic) heterogeneity.

On line 255 in the discussion, I finally saw something I could relate to: “Migratory species would not be directly subject to energy bottlenecks.” But this brings up (to me, at least, not a bird expert), why sample only in May and June? I understand that in our locale (Glen Helen Ecological Institute), bird counts are always made at the same time of year. But from the advertisements regarding participation in the bird count, these months are supposed to pick up migrating birds. I did not see how the author might have separated such bids from those that do not migrate (or whether this was attempted). But if such a distinction is made, then the results either become species dependent (and less general) or one could simply distinguish between birds that are only seen in these months and those that are seen all year.

I saw no mention of John Harte’s results relating analytically species richness to spatial scale. Is there a power-law relationship also for aves? Is it expected?

The author suggests at the end that an important additional input could be climate. Mean climate variables P, PET, and ET are already (to varying degrees individually) strongly correlated with species richness (Currie, 1991) and with productivity (Rosenzweig, 1968). Variability in climate variables (e.g., - I think – bottlenecks, not seasonality) was also addressed in Currie (1991).

I am very much aware of my own limitations in this field and would not wish to suggest that the author is alone at fault for the difficulty in communication. However, if some of these issues are also brought up by referees who are more closely related, then the author should probably consider revising accordingly. For overall understanding, the figures do help and are useful, although I only pick up the areas where the specific predictors are useful. Understanding how the scale-dependence affects these conclusions was not so easy.

- Currie, D. J. (1991). Energy and large-scale patterns of animal and plant species richness. The American Naturalist, 137(1), 27-49.
- Rosenzweig, M. L. (1968). Net primary productivity of terrestrial communities: prediction from climatological data. The American Naturalist, 102(923), 67-74.
- Harte, J., Smith, A. B., & Storch, D. (2009). Biodiversity scales from plots to biomes with a universal species–area curve. Ecology Letters, 12(8), 789-797.

---

## Round 0.2 · accepted · Accept

I have reviewed the revised manuscript and appreciated the author's responses to reviewer suggestions. I am happy with the current version. It is ready for publication.